# The Role of Hormonal Replacement Therapy in BRCA Mutated Patients: Lights and Shadows

**DOI:** 10.3390/ijms24010764

**Published:** 2023-01-01

**Authors:** Vera Loizzi, Miriam Dellino, Marco Cerbone, Francesca Arezzo, Gerardo Cazzato, Gianluca Raffaello Damiani, Vincenzo Pinto, Erica Silvestris, Anila Kardhashi, Ettore Cicinelli, Eliano Cascardi, Gennaro Cormio

**Affiliations:** 1Oncology Unit IRCSS Istituto Tumori “Giovanni Paolo II”, Department of Interdisciplinary Medicine (DIM), University of Bari “Aldo Moro”, 70124 Bari, Italy; 2Obstetrics and Gynecology Unit, Department of Biomedical Sciences and Human Oncology, University of “Aldo Moro”, 70124 Bari, Italy; 3Section of Pathology, Department of Emergency and organ transplantation (DETO), University of Bari “Aldo Moro”, 70124 Bari, Italy; 4Gynecologic Oncology Unit, IRCCS Istituto Tumori Giovanni Paolo II, 70124 Bari, Italy; 5Pathology Unit, FPO-IRCCS Candiolo Cancer Institute, 10060 Candiolo, Italy; 6Department of Medical Sciences, University of Turin, 10124 Turin, Italy

**Keywords:** BRCA, HRT, RRSO, menopausal syndrome, BRCA1, BRCA2, hormonal deprivation, endometrial cancer, breast cancer

## Abstract

**Simple Summary:**

Advances in molecular genetics have radically changed all aspects of cancer prevention, screening and treatment and, today, women who are suspected of being at risk of hereditary cancer should always undergo genetic counselling. Furthermore, today there is growing attention on the tumor suppressor genes BRCA 1 and BRCA2. In addition, there is growing attention on women who carry mutations of the tumor suppressor genes BRCA 1 and BRCA2 and undergo a prophylactic risk-reducing salpingo-oophorectomy. This paper explores the pathobiology of BRCA1\2 cancer genes in a translational perspective, focusing on molecular aspects of hormonal therapy in early menopausal women as well as on the latest evidence-based guidelines for clinicians. Physicians should counsel motivated patients who ask for relief from early menopausal symptoms about the risks and benefits of hormonal therapy, as well as tailoring and appropriate treatment for selected patients.

**Abstract:**

All cancers develop as a result of mutations in genes. DNA damage induces genomic instability and subsequently increases susceptibility to tumorigenesis. Women who carry mutations of BRCA 1 and BRCA2 genes have an augmented risk of breast and ovarian cancer and a markedly augmented probability of dying because of cancer compared to the general population. As a result, international guidelines recommend that all BRCA1\2 mutation carriers be offered risk-reducing bilateral salpingo-oophorectomy at an early age to reduce the risk of cancer and decrease the mortality rate of this high-risk population. NCCN guidelines recommend risk-reducing bilateral salpingo-oophorectomy in pre-menopausal women, between 35–40 years in BRCA1 mutation carriers and between 40–45 years in BRCA2 mutation carriers. Unfortunately, the well-documented reduction of cancer risk is counterbalanced by early sterility and premature ovarian failure with an early onset of secondary menopausal syndromes such as neuromotor, cardiovascular, cognitive and urogenital deficiency. Hormonal replacement therapy significantly compensates for hormonal deprivation and counteracts menopausal syndrome morbidity and mortality; however, some data suggest a possible correlation between hormonal medications and cancer risk, especially in BRCA1\2 carriers who undergo long-term regimens. Conversely, short-term treatment before the age of natural menopause does not appear to increase the cancer risk in BRCA1 mutation carriers without a personal history of breast cancer after prophylactic surgery. Few data are available on BRCA2 mutation carriers and more well-designed studies are needed. In conclusion, clinicians should propose short-term hormone replacement therapy to BRCA 1 carriers to counteract hormonal deprivation; personalized counselling should be offered to BRCA2 mutation carriers for a balance between the risks and benefits of the treatment.

## 1. Pathobiology and Clinical Aspects of BRCA1\2 Cancer Genes 

Breast-cancer genes 1 and 2 (BRCA 1, BRCA2) encode for proteins involved in tumor suppression. Women carriers of BRCA1\2 mutations have a deficiency in the DNA-repair mechanism and dysregulation in the cell cycle. DNA damage induces genomic instability and subsequently increases susceptibility to tumorigenesis, increasing ovarian cancer (OC) and breast cancer (BC) risk and worsening overall survival (OS) compared to the BRCA wild-type population [1]. The BRCA1 gene, located on chromosome 17, is involved in cell-cycle checkpoints in response to DNA damage and DNA repair. BRCA1 mutation carriers have a 44% risk of OC and 72% risk of BC compared to the general population. The BRCA2 gene, located on chromosome 13, is involved in the repair of double-strand DNA breaks. BRCA2 mutation carriers have a 17% risk of OC and 69% risk of BC compared to the general population [2,3]. Moreover, BRCA1\2 mutations increase the risk of aggressive histotypes such as ER-\PR- negative and HER2-negative (i.e., “triple negative” BC and serous adenocarcinoma\high grade OC) [4]. For women with an inherited germline pathogenic variant in an ovarian cancer risk gene, there is a 50% chance of passing the risk on to each of their children. Ovarian cancer causes more deaths than any other cancer of the female reproductive system. The high mortality rate is caused by asymptomatic growth of the tumor and delayed onset of symptoms. Because of the absence of reliable methods of early detection, OC is often diagnosed in advanced stages with a poor prognosis [5]. While screening for ovarian cancer has not been shown to improve outcomes, risk-reducing bilateral salpingo-oophorectomy (RRSO) is the most efficient prophylactic strategy proposed by international guidelines for ovarian cancer prevention in BRCA1\2 mutation carriers. 

International guidelines recommend RRSO after completion of childbearing or at 35–40 years of age for BRCA1 mutations carriers or at 40–55 years of age for BRCA2 mutation carriers to reduce the risk of breast and ovarian cancers. Data from several studies indicate that RRSO is associated with a decrease in OC (96%) and BC (50%) occurrence [6] and a significant reduction in the all-cause mortality rate, especially in BRCA1 mutation carriers (HR 0.45, *p* < 0.0001) [7]. In BRCA2 mutation carriers, the data were not statistically significant (HR 0.88, *p* = 0.75) and more studies are needed [8,9,10,11]. 

Approximately 65% of women carrying the BRCA1 mutation will have RRSO before their primary menopause and will experience secondary menopause and premature ovarian failure (POF) with an early deficiency in sex hormones. Hormonal deficiency causes a metabolic and endocrine imbalance that induces cardiovascular and urogenital syndromes, reduces bone density and induces cognitive impairment and dementia. Women experiencing early menopause have an increased risk of diabetes and a difference of −3.1 (95% CI, −5.1 to −1.1) years overall in life expectancy compared with the age at normal menopause [12].

Several non-hormonal and hormonal treatments have been proposed in European Society of Human Reproduction and Embryology (ESHRE) guidelines to decrease the severity of menopausal symptoms and, in turn, increase the quality of life. Behavioural changes that can benefit hormonal assessment are giving up smoking, doing regular exercise and having a regular diet. Hormonal replacement therapy (HRT) may play a leading role in the primary prevention of cardiovascular disease and osteoporosis. Benefits have also been described on cognitive, sexual and urological issues. There is reassuring data on the use of short-term HRT in BRCA1\2 mutation carriers without a personal history of breast cancer after prophylactic bilateral salpingo-oophorectomy. Use of short-term HRT after oophorectomy was associated with relatively small changes in life expectancy but better outcomes on quality-of-life issues [13]. According to the ESHRE guidelines, 17-beta-estradiol or conjugated equine estrogens should be the medications of choice, in combination with oral cyclical progesterone in women with an intact uterus to protect the endometrium. HRT should be initiated at the time of diagnosis and should be discontinued before 52 years of age. An annual clinical follow-up should be performed with attention to compliance [14].

## 2. Molecular Aspects of HRT

Menopause is the most influential biological, biochemical and health-related event for women that can occur spontaneously (primary menopause) or can be surgically induced by oophorectomy. Estrogens interact with neuronal networks at many levels and affect brain development and aging [15]. The modulation by estrogen may therefore account for hormonal regulation of cognition and mood, as for instance women have an increased risk of developing Alzheimer’s dementia and major depression compared to men, but the pathophysiological mechanisms behind these differences are unknown [16]. The Mayo Clinic Cohort of Oophorectomy and Aging followed 2390 women for a 25–30-year period, finding that women with a bilateral/unilateral oophorectomy before primary menopause had an increased risk of dementia, cognitive impairment and parkinsonism, regardless of the indication for oophorectomy The risk increased with a younger age at the time of the oophorectomy. The Religious Orders Study and Memory and Aging Project followed 1884 women over a period of 18 years, finding that a younger age at the time of BSO was associated with a rapid decline in global cognitive function and a high risk of Alzheimer’s disease. The initiation of HRT in the first 5 years after the onset of menopause with prosecution for at least 10 years was associated with improvement in cognitive decline [17]. The widespread distribution of estrogen and progesterone receptors in the brain is responsible for the dramatic changes that the hormonal milieu effect on neurotransmitters and the neuroendocrine system during the menopausal transition period, thus modifying intermediate phenotypic patterns. Neuroscience is investigating the effects of hormonal changes on neuronal activity and the developing of vasomotor symptoms, age-related memory loss and cognitive and emotional complaints that affect quality of life and overall functioning [18]. Studies of neurochemical pathways have pinpointed the cholinergic, serotonergic, and dopaminergic systems as biological mediators of hormonal influences on the brain. Some studies suggest a preservative/stimulating function of estrogen for the cerebral blood flow in several cortical regions.

### 2.1. HRT and Cholinergic System

Estrogen is indeed known to provide neurotrophic support, possibly through synergistic actions with the cholinergic system, and to regulate acetylcholine release. Acetylcholine (ACh) is a neurotransmitter critically involved in the mechanism of memory, learning and attention. Both estrogen receptors have been identified in the nuclei of the human basal forebrain, which is the source of major cholinergic innervations to the cerebral cortex, hippocampus, and hypothalamus [19]. The decay of the central cholinergic system is involved in physiological aging. A cross-sectional PET study of women who initiated HRT within 2 years of menopause, and who used it for more than 10 years, indicated a higher cholinergic activity (8–10%), bilaterally in the posterior cingulate of combined estrogen-progestagen users compared to non-users [20]. Thus, while low menopausal estradiol levels are likely to diminish, HRT may enhance the survival or plasticity of cholinergic cells in postmenopausal women. In a study on 30 right-handed young (26–47 years) healthy premenopausal women with benign leiomyomata uteri who were prescribed GnRHagonist as part of their routine clinical management, acute loss of ovarian hormones affected the cholinergic system, producing a significant behavioral deficit in overall memory performance, which is dependent on the fronto-temporal brain regions [21].

### 2.2. HRT and Serotoninergic System

Estrogen, alone or combined with progesterone, modulates the serotonergic function at the level of the neurotransmitter synthesis, turnover, release, and receptor, and has a neuroprotective effect on serotonin neurons [22]. The serotonin system critically modulates brain functions related to both cognition and mood; it is thus plausible to hypothesize a synergistic effect of estrogen and serotonin on cognitive functioning, emotion processing, and affective regulation [22]. Three longitudinal PET studies assessing the serotonin receptor 2A (5-HT 2A) highlighted a general increased HRT-mediated binding potential in frontal and cingulate regions in postmenopausal women compared to the baseline. The mechanism by which HRT leads to a globally distributed 5HT 2A binding increase is not fully understood. Indeed, selective serotonin re-uptake inhibitors are valid alternatives to estradiol for the treatment of vasomotor symptoms, i.e., hot flushes, and depressive symptoms [23,24]. Chronic exposure to 4-vinylcycloxene diepoxide (VCD) in rodents as a model of early menopause in women showed that estradiol therapy may improve perimenopause symptoms by increasing the biosynthesis of progesterone and boosting the serotonin pathway from the caudal dorsal raphe nucleus to the dorsal hippocampus [25].

### 2.3. HRT and Dopaminergic System

Estrogen could play a protective role against Parkinson disease in women. Estrogens, alone and in combination with progesterone, are likely to affect the dopaminergic system too as findings in non-human primates suggest an effect, although scattered and less prominent, of estrogen on the dopaminergic system [26]. Estrogen was found to modulate nigrostriatal dopaminergic activity through increased dopamine synthesis [27] and decreased oxidative stress, protecting dopaminergic neurons against apoptosis [28]. Oophorectomy before menopause was shown to increase the risk of parkinsonism by 68% [29] HRT has been associated with reduced PD risk in women with an early menopause [30], consistent with the prevailing theory of a protective effect of estrogens.

### 2.4. HRT and Neurogenesis

Of special interest in relation to neurogenesis and synaptic plasticity is the hippocampus, a crucial region in memory and learning [19]. A larger hippocampal volume in HRT users has been indicated by magnetic resonance imaging studies, with the first evidence reported by Eberling et al. (2003) [31].

### 2.5. HRT and Psychiatric Disorders

The peri menopause is often accompanied by symptoms of depression and anxiety [23]. HRT has been associated with decreased depressive symptoms in menopausal women [32], although it depends on psychiatric history, age and study design; while other studies, such as the Women’s Health Initiative study, provided conflicting evidence [16]. Moreover, free estradiol levels were positively correlated with mood in a recent large study of healthy women during early postmenopause (Henderson et al., 2013), but again HRT, in particular current use, was associated with symptoms of depression and anxiety in two large cohorts of peri-menopausal women [33]; however, estradiol is generally recognized to have positive effects on mood [34].

## 3. Lights and Shadows of HRT

### 3.1. Brief History of HRT in Primary Menopausal Women

HRT has been used for more than 70 years, yet there are still disagreements regarding its clinical value and its safety in BRCA1\2 carriers who underwent early menopause after RRBSO. Data from the Women’s Health Initiative (WHI) trial and Million Women Study (MWS) suggested a link between HRT and cancer risk in women (Table 1) who underwent primary menopause [35]. 

The first results of the WHI (2002) after a mean follow-up period of 5.2 years showed an increased incidence of coronary events, stroke, pulmonary embolism and breast cancer in women who received a combination of estrogen with progestogen. The WHI study was prematurely halted because of evidence of an increased risk of breast cancer, with no apparent beneficial cardiovascular effects. Given these results, it seemed that HRT risks outweigh benefits and the message was that HRT, with no specification of type and route of administration and no distinction between users and their age, was dangerous [44]. The follow-up of the WHI continued for 13 years and an age stratification of the cardiovascular outcomes showed the beneficial effects of HRT on the cardiovascular system, coronary diseases and all-cause mortality in women between 50 and 59 years and women within 10 years of primary menopausal onset [45]. The Million Women Study, a large observational study by an Oxford group of epidemiologists, reported an elevated risk of breast cancer among users of HRT irrespective of formulation, dosage, route of administration and regimen. Current users of HRT at recruitment were more likely than those who had never used it to develop breast cancer (adjusted relative risk 1.66 [95% CI 1.58–1.75], *p* < 0.0001) and die from it (1.22 [1.00–1.48], *p* = 0.05). The risk was higher for women using combined oestro-progestin (RR 2.00 [1.88–2.12], *p* < 0.0001) [46]. The study has been widely criticized on several methodological issues such as selection bias and detection bias. There were also misclassifications of the time, type, and duration of hormonal therapy [47]. The 2016 Revised Global Consensus Statement on Menopausal Hormone Therapy [48] reaffirmed that HRT remains the most effective treatment for vasomotor symptoms and significantly lowers the risk of osteoporosis-related fractures in postmenopausal women. HRT is effective in vulvovaginal atrophy and may improve sexual function and other related symptoms such as joint and muscle pains, mood changes and sleep disturbances. In addition, the statement confirms that women experiencing a spontaneous or iatrogenic menopause before the age of 45 years and particularly before 40 years are at a higher risk of cardiovascular disease and osteoporosis and may be at increased risk of affective disorders and dementia. In such women, HRT reduces symptoms and preserves bone density. Observational studies that suggest HRT is associated with a reduced risk of heart disease, longer lifespan, and reduced risk of dementia require confirmation in RCTs. 

### 3.2. HRT and Menopausal Symptoms in BRCA1\2 Carriers

After a RRSO at a premenopausal age, women experience more severe menopausal symptoms compared to naturally menopausal women. The loss of ovarian hormone synthesis after oophorectomy before menopause induces psychological and physical adjustment and impairs the quality of life and sexual functioning [49]. Half of the women reported severe urogenital symptoms. Psychological symptoms tended to decrease with time, as women reported more severe symptoms directly after RRSO compared to ≥10 years after RRSO [49]. Stuursma et al. [50], in a recent systematic review, reported that estradiol seems to have a beneficial effect on a depressed mood in the short term 3–6 years after surgery or 2 years (median) after surgery with high heterogeneity (Standardized mean difference SMD: −1.37, 95% CI: (−2.38–−0.37) *p* = 0.007). Testosterone had a beneficial effect on overall sexual functioning in the short to medium term 4.6 years after surgery (SMD 0.38, 95% CI (0.11–0.65) and on sexual desire in the medium term at least 3–12 months after surgery (SMD 0.38, 95% CI (0.19–0.56). In a recent psycho-oncology survey, Grandi et al. [51] demonstrated that BRCA mutation carriers are likely to overstate the negative effects of HRT and underestimate the benefits. It is essential to raise awareness regarding menopausal symptoms and management options.

### 3.3. HRT and Cancer Risk in BRCA1\2 carriers

BRCA1\2 mutation carriers undergoing RRSO have different risk factors from women who experience primary menopause: the genetic susceptibility to cancer is mitigated by a younger age at the onset of hormone deficiency, a shorter hormone exposure time and an increased number of clinical evaluations because of screening. In the cohort of women with BRCA1/2 mutations, short-term HRT following RRSO did not alter the significant reduction in breast cancer risk associated with RRSO during the available follow-up period. Huber at al. (2020) reviewed systematically the literature about BRCA1\2 mutation carriers undergoing HRT [36]. (Table 1) In BRCA carriers, RRSO has been shown to decrease gynecologic cancer-specific, as well as overall, mortality [52]. The data suggested that the reduction in breast cancer risk associated with RRSO was not different in women who had taken HRT (HR 0.37; 95% CI, 0.14 to 0.96) from what it was in the overall cohort [10,36,53]. HRT use among RRSO patients did not significantly alter postsurgical breast cancer risk (HR 1.35; 95% CI, 0.16 to 11.58), including in BRCA1 mutation carriers [54,55,56,57]. Moreover, a possible protective effect of HRT on BC in BRCA mutation carriers has been reported, with a reduction of 8% of cancer risk for every year of an estrogens-only regimen and an increase of 8% of cancer risk for every year of progestin-only regimen. A subgroup analysis carried out in Italy including women before the age of 45 undergoing RRSO observed a 18% reduction in BC risk for every year of estrogen-only replacement and a non-significant increase in BC risk of 14% for every year of a combined oestro-progestin regimen [58]. A recent meta-analysis of a total of 1100 patients concluded that BC risk associated with HRT was similar in the entire population with no statistical differences in cancer frequency between BRCA-mutated and BRCA wild-type women. A subgroup analysis reported no statistically significant differences in BC risk comparing women who used the estrogen regimen on its own and women who used combined oestro-progestins formulations. However, several Italian researchers observed a positive trend of a lower, albeit insignificant, risk of BC in those who received an estrogen-only regimen compared to those who received estrogen and progesterone.

A controversial field of study is the choice to perform a concurrent hysterectomy (CH) at the same time as a RRSO to reduce the risk of endometrial cancer (EC). A meta-analysis of 13,871 carriers of BRCA1 and BRCA2 mutations demonstrated pooled prevalence rates of EC in BRCA1\2 mutation carriers of 0.59%. The EC prevalence was 0.62% in BRCA1 mutation carriers and 0.47% in BRCA2 mutation carriers [59]. The studies included in this meta-analysis suggested a slightly increased risk of endometrial cancer in BRCA mutation carriers (RR 1.18 (95% CI, 0.7–2.0), mainly BRCA1.). Evidence has established that the use of unopposed estrogen increases the risk of EC; recent data resulting from a nationwide Italian survey reported an augmented risk of aggressive EC histotypes such as serous and serous-like in BRCA1 carriers undergoing RRSO without CH [60] and an increased risk of EC among women BRCA1 carriers who use estrogens as HRT [56]. However, NCCN guidelines do not recommend routine CH with RRSO because the risk of surgery outweighs the benefits [43]. For this reason, clinicians should counsel women undergoing RRSO and propose a concomitant hysterectomy only in selected cases with multiple risk factors for EC and in particular conditions in which benefits outweigh the risk and morbidity of the surgery [61]. Despite a lack of official recommendations for CH among BRCA1 carriers, CH for uterine cancer risk-reduction is becoming more common over time. In a 2019 paper by S. Gordhandas et al. [62] CH was more common among women with BRCA1 mutations vs. BRCA2 mutations (31% vs. 14%, *p* = 0.02). Uterine cancer risk-reduction was the most common indication for CH (*n* = 22, 58%). Following the 2016 publication, CH was significantly more common compared to before, 43% vs. 18%, respectively (*p* = 0.006).

Few data are currently available on HRT in BRCA 2 mutation carriers because of the older age at which these carriers usually undergo RRSO and the tendency to develop hormone receptor-positive BC. Sample sizes were often small, especially regarding BRCA2 mutation carriers who were included in only a few studies. In addition, most of the studies did not perform subgroup analyses for BRCA2. For these reasons, in this sub-group of patients, HRT should be administered with caution and after an evaluation of risk factors.

## 4. Perspectives and Conclusions

Women with BRCA1 and BRCA2 mutations have an increased risk of OC and BC throughout life compared to the general population and should therefore undergo RRSO to decrease the cancer risk. HRT following RRSO in BRCA1/2 mutation carriers does not seem to have an adverse effect or negate the reduction in breast cancer risk associated with RRSO. Adequate HRT after RRSO should be offered to premenopausal women to avoid postmenopausal symptoms and chronic diseases resulting from low estrogen levels such as osteoporosis and myocardial infarction. Regarding BRCA1/2 mutation carriers who have not undergone RRSO, HRT does not seem to have a relevant impact on cancer risk. Short-term HRT may be considered in BRCA1/2 mutation carriers without RRSO to reduce perimenopausal symptoms. In the case of BRCA 1 mutation carriers, clinicians should be aware of recent evidence that reassures that short-term HRT does not increase BC risk. In women who receive a concomitant hysterectomy, estrogen-only HRT is the safest and most reasonable choice due to possible increases in endometrial cancer risk. However, indications should be strict and patients should be informed about limited data on the safety concerning cancer risk. Nevertheless, evidence is limited and further studies with a large prospective and randomized cohorts are necessary. Future researchers should also evaluate the role of hysterectomy as a risk-reducing surgery in BRCA mutation carriers with concurrent risk factors for EC. HRT’s goal is to prevent menopausal symptoms and cardiovascular diseases, osteoporosis, and long-term morbidity associated with hormone depletion. Physicians should publish all the available information regarding the use of HRT in women with BRCA 1\2 mutations during the counselling of RRSO. The decision to undergo HRT is a complex one and should be made in consultation with a gynecologic oncologist. In addition, the patient’s knowledge of the benefits, risks and limitations of HRT should be assessed as well as the patient’s goals. Topics that should be assessed include impairments to reproduction, risk assessment of premature menopause and other medical issues. 

## Figures and Tables

**Table 1 ijms-24-00764-t001:** Most relevant study on HRT in BRCA mutation carriers and risk of ovarian, endometrial, and breast cancer. Revised and modified from Huber et al. (2020) [36].

Study on Ovarian Cancer	Study Design	Number	Results	
**Kotsopoulos et al. (2006)** [37]	Case–control study	Cases: 117 *BRCA1*, 45 *BRCA2*	No increase in risk of OC in *BRCA1* and *BRCA2* carriers	*BRCA1*: OR = 0.92; 95% CI 0.50–1.70; *p* = 0.80
		Controls: 256 *BRCA1*, 119 *BRCA2*		*BRCA2*: OR = 0.89; 95% CI 0.29–2.39; *p* = 0.74
**Perri et al. (2015)** [38]	Retrospective cohort study	718 *BRCA1*, 331 *BRCA2* (use of HRT: 105)	Increase in risk of OC *in BRCA1* and *BRCA2* carriers	*BRCA1*: OR = 1.
				66; 95% CI 0.89–3.08; *p* < 0.001
				*BRCA2*: OR = 3.04; 95% CI 1.19–7.8; *p* < 0.001
**Study on Breast Cancer**	**Study Design**	**Number**	**Results**	
**Eisen et al. (2008)** [39]	Case–control study on *BRCA1* mutation carriers with and without RRSO	236 case–control pairs *BRCA1*	Decrease in risk of BC in *BRCA1* carriers	OR = 0.58; 95% CI 0.35–0.96;
				*p* = 0.03
**Kotsopoulos et al. (2016)** [40]	Case–control study on *BRCA1* mutation carriers with and without RRSO	432 case–control pairs *BRCA1*	No increase in risk of BC in *BRCA1* carriers	OR = 0.80; 95% CI 0.55–1.16;
				*p* = 0.24
**Rebbeck et al. (1999)** [7]	Case–control study on *BRCA1* and *BRCA*2 mutation carriers after RRSO	*BRCA1 and BRCA2 mutation carriers after RRSO*	Decrease in BC risk in *BRCA1* and *BRCA2* associated with RRSO after	HR = 0.42; 95% CI = 0.22–0.81
		122 *BRCA1 (*43 cases with RRSO, 79 controls without surgery)	exclusion of HRT users	
**Rebbeck et al. (2005)** [10]	Prospective cohort study	315 *BRCA1* (110 with RRSO),	No increase in BC risk associated with RRSO	HR = 0.37; 95% CI 0.14–0.96
		147 *BRCA2* (45 with RRSO)		
**Domchek et al. (2011)** [41]	Extension and follow-up to Rebbeck et al. (2005)	*BRCA1* mutation carriers with and without RRSO 795 *BRCA1*, 504 *BRCA2* (321 with RRSO)	No increase in BC risk in *BRCA1* and *BRCA2* carriers associated with RRSO	*BRCA1*: decrease in risk for
				women with (HR = 0.52; 95%
				CI 0.30–0.92) and without
				RRSO (HR = 0.29; 95% CI
				0.13–0.69)
				No association with different
				types of HRT
**Kotsopoulos et al. (2018)** [42]	Prospective cohort study	872 *BRCA1* (all with RRSO)	No increase in BC risk in *BRCA1* carriers associated with RRSO	HR = 0.97; 95% CI 0.62–1.52;
				*p* = 0.89
**Study on Endometrial Cancer**	**Study Design**	**Number**	**Results**	
**Segev et al. (2015)** [43]	Case–control study	Cases:	No increase of EC risk in *BRCA1* and *BRCA2* carriers	OR = 0.73; 95% CI 0.33–1.63; *p* = 0.44
		62 *BRCA1*, 21 *BRCA2*		
		Controls:		
		951 *BRCA1*, 76 *BRCA1*		
**Study on Ovarian Cancer**	**Study Design**	**Number**	**Results**	
**Kotsopoulos et al. (2006)** [37]	Case–control study	Cases: 117 *BRCA1*, 45 *BRCA2*	No increase in risk of OC in *BRCA1* and *BRCA2* carriers	*BRCA1*: OR = 0.92; 95% CI 0.50–1.70; *p* = 0.80
		Controls: 256 *BRCA1*, 119 *BRCA2*		*BRCA2*: OR = 0.89; 95% CI 0.29–2.39; *p* = 0.74
**Perri et al. (2015)** [38]	Retrospective cohort study	718 *BRCA1*, 331 *BRCA2* (use of HRT: 105)	Increase in risk of OC in *BRCA1* and *BRCA2* carriers	*BRCA1*: OR = 1.
				66; 95% CI 0.89–3.08; *p* < 0.001
				*BRCA2*: OR = 3.04; 95% CI 1.19–7.8; *p* < 0.001
**Study on Breast Cancer**	**Study Design**	**Number**	**Results**	
**Eisen et al. (2008)** [39]	Case–control study on *BRCA1* mutation carriers with and without *RRSO*	236 case–control pairs *BRCA1*	Decrease in risk of BC in *BRCA1* carriers	OR = 0.58; 95% CI 0.35–0.96;
				*p* = 0.03
**Kotsopoulos et al. (2016)** [40]	Case–control study on *BRCA1* mutation carriers with and without *RRSO*	432 case–control pairs *BRCA1*	No increase in risk of BC in *BRCA1* carriers	OR = 0.80; 95% CI 0.55–1.16;
				*p* = 0.24
**Rebbeck et al. (1999)** [7]	Case–control study on *BRCA1* and *BRCA/2* mutation carriers after RRSO	*BRCA1 and BRCA2 mutation carriers after RRSO*	Decrease in BC risk in *BRCA1* and *BRCA2* associated with RRSO after	HR = 0.42; 95% CI = 0.22–0.81
		122 *BRCA1 (*43 cases with RRSO, 79 controls without surgery)	exclusion of HRT users	
**Rebbeck et al. (2005)** [10]	Prospective cohort study	315 *BRCA1* (110 with RRSO),	No increase in BC risk associated with RRSO	HR = 0.37; 95% CI 0.14–0.96
		147 *BRCA2* (45 with RRSO)		
**Domchek et al. (2011)** [41]	Extension and follow-up to Rebbeck et al. (2005)	BRCA1 mutation carriers with and without RRSO 795 *BRCA1*, 504 *BRCA2* (321 with RRSO)	No increase in BC risk in *BRCA1* and BRCA2 carriers associated with RRSO	*BRCA1*: decrease in risk for
				women with (HR = 0.52; 95%
				CI 0.30–0.92) and without
				RRSO (HR = 0.29; 95% CI
				0.13–0.69)
				No association with different
				types of HRT
**Kotsopoulos et al. (2018)** [42]	Prospective cohort study	872 *BRCA1* (all with RRSO)	No increase in BC risk in BRCA1 carriers associated with RRSO	HR = 0.97; 95% CI 0.62–1.52;
				*p* = 0.89
**Study on Endometrial Cancer**	**Study Design**	**Number**	**Results**	
**Segev et al. (2015)** [43]	Case–control study	Cases:	No increase of EC risk in BRCA1 and BRCA2 carriers	OR = 0.73; 95% CI 0.33–1.63; *p* = 0.44
		62 *BRCA1*, 21 *BRCA2*		
		Controls:		
		951 *BRCA1*, 76 *BRCA1*		

## Data Availability

Not applicable.

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
