# Peer review of "The Role of Hormonal Replacement Therapy in BRCA Mutated Patients: Lights and Shadows"

_ijms, 2023, doi:10.3390/ijms24010764_

Round 1

Reviewer 1 Report

This review addresses an extremely contemporary and interesting topic, that being the use of hormone replacement therapy (HRT) in BRCA mutation carriers. However, I believe that the central theme in the title has not been addressed thoroughly, and a lot of non-pertinent information has been added to the work.

In the following paragraphs a more specific list of corrections and clarifications related to this work:

-       The simple summary is not representative of the theme of the article. It does not clarify the key points of the paper.

-       Abstract

o   Line 36: international guidelines recommend risk-reducing salpingo-oophorectomy between 40-45 years in BRCA2 mutation carriers.

-       1. Pathobiology of BRCA1\2 cancer genes

o   Reference [1] is about BRCA radiosensitivity and does not seem appropriate for the concept being expressed in the previous sentence.

o   Line 66: “germline pathogenic variant in an ovarian cancer risk gene, there is a 50% chance of passing the risk on to each of their children.” Referring exclusively to a mutation that increases the risk of ovarian cancer can be confusing.

o   Lines 81-82 are a repetition of lines 75-76

o   Lines 80-86: The guidelines agree on the age at which RRSO should be recommended, merge the sentences referring to the guideline indications into a single sentence.

o   Line 87: why is this information related only to BRCA1 patients?

o   Lines 94-108: being the section titled “Pathobiology of BRCA1\2 cancer genes” this paragraph is not relevant

o   Line 106: HTR seems incorrect

-       2. Molecular aspects of HRT

o   this section would be appropriate and interesting if it were to investigate the molecular mechanisms related to sudden and early menopause post-RRSO. The entire section delves into the pathogenetic mechanisms of spontaneous menopause with gradual hormone deprivation, i.e “the overall level of oestrogen to which 119 the female brain is exposed fluctuates and gradually declines”.

o   The paragraph could be interesting but is too in-depth on issues that are not directly related to the title, it is taken out of context in a review of the literature related to HRT in BRCA mutation carriers

o   The paragraph seems very unbalanced compared to the little that is later said about HRT and BRCA

-       3. Lights and Shadows of HRT

o   The table is not clear and lacks the required footer

o   The section is written in a poorly structured manner, with few headings, and with one sentence following another loosely.

o Lines 200-267: the history of HRT and the main studies on the topic should not be explored so thoroughly in such a specific review, it does not bring anything new to the literature. Many of the sections refer to studies on menopause in general, they do not add any news related to the use of HRT in the specific subpopulation of BRCA mutation carriers.

o   238-242: this is relevant to the topic

o   Ospemifene is mentioned but not adequately explored and it is not contextualized concerning the population of interest in the study

o   268-291, this is the only passage where HRT and BRCA mutations are discussed. it is too superficial a coverage, considering that it is the title of the review

o  Lines 292-303: the discussion of whether hysterectomy should be performed in patients with BRCA1 mutation looks out of context. Moreover, this controversial subject should be supported with some data.

-       4. Perspectives and Conclusions

o   Line 314 RRBSO Is an acronym that has never been used before in the text

Author Response

Thanks for the valuable suggestions; the answers are attached

Reviewer 2 Report

There is too much detailed information about MHT en general population page 6), it should be more focused in the topic in study.

Page 6 needs to be more "reading friendly", using separation points and different paragraphs.

 References 45, 46 and 47 don't seem related to the topic.

I think the paper should be re ordered by the authors giving less information about MHT in general population, and trying to be more specific about it in BRCA mutated patients.

Author Response

(The authors gave the same response as above.)

Round 2

Reviewer 1 Report

Reviews have been addressed